# ON INHERENT LIMITATIONS OF GPT/LLM ARCHITECTURE

## ABSTRACT

In this paper, we show that reasoning/proving issues of GPT/LLM are an inherent logical consequence of the architecture. Namely, they are due to a schema of its prediction mechanism of the next token in a sequence, and randomization involved into the process. After natural formalization of the problem into a domain of finite graphs, $G(\omega)$, we prove the following general theorem:

*For almost all proofs, any learning algorithm of inference, that uses randomization in $G(\omega)$, and necessitates veracity of inference, is almost surely literal learning.*

In the context, "literal learning" stands for one which is either vacuous, i.e. $\forall x \, [P(x) \implies Q(x)]$ where $P(x)$ is false for every $x$, or create a random inference from a false assumption (hallucination), or it essentially memorizes the inferences from training/synthetic data.

## 1 NOTATION AND TERMS

GPT/LLM stands for algorithmic representation of transformer with attention viewed as a main inference mechanism for $LLM$.

$G(\omega)$ stands for the infinite set of finite graphs, and the first-order model on the set as a domain, where the connectivity of nodes $n_1 \sim n_2$ stands for the first countable ordinal.

Formal definitions for the language of the first-order theory can be found in Appendix B. It contains all the necessary information on the 0-1 law.

In this setting, a fault in $G(\omega)$ is an erroneous proof – that is, a chain of thought containing a false implication ($n_1 \not\sim n_2$), or a false assumption (node $n_1$ represents falsehood).

"Randomization" on a probability space means that the events inference mechanism admits variability in the next token selection (such as random seed initialization, temperature, beams search, etc.).

A "Learning algorithm" is a machine learning algorithm that, after being trained on data, produces output based on the training.

"Literal learning" stands for one which is either vacuous, i.e. $\forall x \, [P(x) \implies Q(x)]$ where $P(x)$ is false for every $x$ or creates a random inference from a false assumption (hallucination), or it essentially memorizes the inferences from training/synthetic data.

First-order logic terms can be found in B.

0-1 Law for graphs is in B.1.

## 2 INTRODUCTION

On one hand, GPT architecture for LLMs demonstrated significant progress in a generative manifestation of summarizations, chat, and representations of materials. On the other hand, the architecture displayed multiple negative effects such as hallucinations, falsehoods, degrading generalization, performance degradation, and alike (e.g., (Yadlowsky et al., 2023a)). In rigorous contexts (where one

requires a consistent mathematical reasoning or a formal proof), the results are consistently discomforting ((Chen et al., 2023), (Hagendorff et al., 2022), (Dziri & et. al., 2023)).

Recently a few authors pointed out various limitations (cf. e.g., (Liu et al., 2023), (Mikhaylovskiy & Churilov, 2023), and (Asher et al., 2023a)). Nonetheless, there have been suggested possible remedies ((Sel et al., 2023), (L. et al., 2023), and (Z. et al., 2022)).

In this paper, we show that these issues are an inherent *logical* consequence of the GPT architecture. As a result, multiple phenomena of transformer inference limitations can be explained from purely logical view; in particular, some results of (Dziri et al., 2023) can be obtained that way. It is shown that some limitations addressed in the paper (e.g., problem of increasingly large parallelism requirement) can be relieved with changing a type of attention.

In general, it appears that there is a latent belief in contemporary literature that *all* limitations of technology can be resolved within the governing transformers' model. The goal of this paper is to prove that the architecture is inherently limited in case of inference that required rigor; thus, these imitations are fundamental and innate.

A crucial observation is a scheme of transformer prediction mechanism of next token in a sequence. Using a natural formalization of the problem into the domain of (standard) finite graphs $G(\omega)$, we prove the following theorem:

*For almost all proofs, any learning algorithm of inference, that uses randomization in $G(\omega)$ [1], and necessitates veracity of inference, is almost surely literal learning.*

We provide a few proofs for this statement. For a quick look at a formal proof, refer to C.2. To develop an intuition for the phenomenon, there is an informal proof (1) where consideration is around 0-1 law.

In this form of the inherent limitation, there are a few basic assumptions that need to be addressed. Namely, we work in the first-order model $G(\omega)$ (appendix B.1), where connectivity between nodes $a$ and $b$, expressible by the first order formula, represents the validity of the implication $a \implies b$ (we are going to use $a \rightarrow b$ for the implication as well, interchangeably). For our purposes, the graph do not have to be directed since only a countable enumeration of the nodes is necessary. Moreover, any algorithmic randomization on the nodes allows us to view $G(\omega)$ as a suitable probability space (C.2). Finally, literal (or vacuous) learning is defined as such that, almost surely, the proof chain, generated by an algorithm, is equivalent (in $G(\omega)$) to one in a training dataset.

A few corollaries follow. For instance, if its formulation is somewhat original, it is easy to notice the issue of solving mathematical problems with LLMs in the case of even low complexity task. Since its solution is unlikely to be found in a holistic form in a training dataset, a correct proof is not to be expected.

It is because, in a rigorous context, GPT has exponentially decreasing odds of finding a valid proof of the result unless it simply "repeats" a known proof, perhaps with trivial modifications (Corollary 3). Another corollary is that degradation of performance is of exponential rate by length of a proof. In other words, an attempt to prove a complex enough statement virtually has no chance of being successful.

In a rigorous context of generating a proof, GPT virtually has no chance to find a valid proof of the result unless it simply "repeats" a known proof, perhaps with trivial modifications (Corollary 3). Another corollary is that the degradation of performance has an exponential rate by the length of a proof. In other words, an attempt to prove a complex enough statement with GPT/LLM has virtually no chance to be successful.

In a novel rigorous context (i.e., when GPT-based architecture is looking to prove a new result, for instance, a hypothesis), that is virtually impossible even for a long enough fragment of the proof. The probability of success becomes infinitesimal quickly for either a fragment of possible proof or a weaker non-trivial statement. That also was empirically shown for data mixtures in (Yadlowsky et al., 2023b).

This has been recently confirmed experimentally in ((Hubinger et al., 2024)). Moreover, the logical view approach enables us to discover the same limitation patterns for LLM and auto-regressive next-

---

[1]defined in B.1

token predictors even though the latter are universal ((Malach, 2023)).

The logical view approach is more effective in generalization by varying a domain obeying the generic 0-1 law.

For instance, the results in ((Dziri et al., 2023)) can be obtained as a partial case in the proof of our main result. The 0-1 law variant for polynomial decision problems ((Blass et al., 1998)) is used for a more instructive proof.

Tools like FunSearch ((Romera-Paredes et al., 2023)) contribute to searching for solution specifications, instead of providing an actual inference.

Recent improvements in LLMs such as a 1 million-long context window would make vacuous inferences quite probable in a standard context, with tantamount consequences to the dangling pointers in software. In the novel context, it is almost certain that the resulting proof will be incorrect, present a hallucination, or both. Thus, the expectations that ChatGPT-4.0 or a similar model (e.g., the agents) would soon be able to reason and plan like a person seem unfounded.

For the alignment problem, we only note that our approach is fundamentally different from that of (Wolf & et. al., 2023) since our methodology is ultimately based on considerations within the first-order logic of appropriate Random Graph theory while theirs is purely statistical.

Note also, that in terms of paper (Nasr & et. al., 2023), leaving the adversarial context of it, we essentially proved that in a rigorous context, given sufficient complexity, LLMs are able only to memorize the existing proofs in the training dataset. Thus, one cannot expect these models to produce a novel non-trivial proof. In terms of (Nasr & et. al., 2023), discoverable and extractable memorizations coincide, given sufficient complexity of a statement $P$.

In other words, given sufficient complexity of a statement $P$, the prompt "Please prove statement $P$" would generate a memorized proof if one exists in the training dataset, present an incorrect proof, or hallucinate (cf. (Chen et al., 2023) as in (Asher et al., 2023b), and (Mikhaylovskiy & Churilov, 2023)). Similarly, an attempt to fix the LLM (e.g., GTP-4) bugs with LLM critique technique ((McAleese & et. al., 2024)) will have only a limited scope of applicability.

In the paper (Dohare et al., 2024), the authors note that the deep-learning system's performance degrades during extended training on new data. A method, proposed in the paper, requires randomization to establish elasticity; so it is likely that not just LLMs but also traditional $FFNs$ admit a version of the main result on inherent limitation. Moreover, an attempt to enrich a model with synthetic data will go only so far as well as an emerging representation of underlying abstraction (cf. (Jin & Rinard, 2024) where the context is not rigorous).

Additionally, the main theorem is a formal statement of statistical nature, applicable to a more general context than GPT/LLM, albeit GPT/LLM is a target example. The future of the GPT/LLM may bring additional mechanisms into discussion, e.g., RAG integration, advanced planning, and domain-specific tuning. One can claim that these additional mechanisms will not be sufficient to overcome the limitations of the core foundation of transformer-based generation for rigorous proving. However, we are not making this claim since the algorithmic representation of these mechanisms may vary on prompt tuning, RL, adapters, and most importantly, RLHF. Thus, post-training techniques are out of the scope of the main result. Note that the result does not contradict Malach (2023) since the scope of the two approaches is entirely different. The formalism of that work is focused on approximating any Turing function while this paper presents a first-order logic view on rigorous reasoning at large.

## 3 MOTIVATION/PRELIMINARIES

To demonstrate the model that proves the inherent limitation phenomena, we need to formalize logically the mechanism employed by GPT/LLM to predict the next token in a sequence. It turns out that randomization used by *GPT* architecture (namely "temperature") is the main reason. However, it isn't easy to devise an alternative when dealing with training on a large text corpus. Note that the architecture becomes too "predictable/plain" if we choose the most probable pattern in the list of candidates for the next token. There, we would need a certain randomization to become "creative".

As we will see, that necessary hack is sufficient to preclude the architecture from ever succeeding in a rigorous context, in a formal setting when we need to infer our next supposition with strict regard to its veracity. A good example would be generating proof for a theorem. Note that, despite infamous issues with Generative Learning with rigor in this context, there have been a few feasible attempts to attack this problem that way (e.g., (Saparov et al., 2023)); moreover, there was a claim that we may be able to "recover" from an ostensibly systematic LLM model faltering in mathematical settings ((Shi et al., 2022)). As is known, these attempts were largely unsuccessful. Our results offer a logical explanation of why.

To that end, we introduce formalism to make the subject rigorous enough to have a logical view. Namely, we present a simple first-order theory on the language of (random) graphs where one can state that the generative inference that admits randomization on implications will almost necessarily lead to logical faults (i.e., with probability 1). This result is based on a 0-1 law (and its variations) in (random) graphs theory.

### 3.1 Defining the context

We can assume that one can enumerate all the inferences using $[n]$, since there is only a countable number of (finite) proofs on a countable number of entities. For a graph in $\mathsf{G}(\omega)$, define property $\mathcal{A} := \{\exists$ nodes $e_1, \ldots e_j$ forming chain $(e_0 \to e_1 \ldots e_i \ldots e_j \to e_m)$ for inference $e_0 \to e_m)\}$. This definition is well-formed since $\mathcal{A}$ is expressed as a first-order sentence in the first-order logic theory for $\mathsf{G}(\omega)$), and the axiom of foundation[2]. For chains above, we need to verify that these are first-order expressions. A suitable framework for this is that of least fixed point extension (cf. (Grohe, 2017)). Namely, if the " $\sim$ " is a connectivity relation, then a chain $C(e_0, e_t)$ where $e_0$ and $e_t$ represent a proof starting node $e_0$ and terminal node $e_t$ respectively, can be expressed as follows:

$$C(e_0, e_t) \leftarrow ((e_0 = e_t) \vee \exists e_i (C(e_0, e_i) \wedge e_i \sim e_t)) \tag{1}$$

Then, by 0-1 Lemma in B.1, we have two possibilities, namely: $p = \lim_{n \to \infty} \mathbb{P}(G_n(\omega) \in \mathcal{A}) = 0$ or it is equal to 1. If it is zero, then no valid proof can be found within the context in the first place. Therefore, $\lim_{n \to \infty} \mathbb{P}(G_n(\omega) \in \mathcal{A}) = 1$. By Lemma 0 in B.1, it follows that $\mathsf{G}(\omega) \models \mathcal{A}$. However, it also means that our inference follows a literal graph representation from the original (i.e., from the given training set). Similar consideration is possible for a novel vs. not novel context. Thus, $p \neq 1$. In this case, we create a hierarchy in $\mathsf{G}(\omega)$ as follows.

Consider chain $(e_0 \xrightarrow{\psi_1} e_1 \ldots e_i \ldots e_j \xrightarrow{\psi_k} e_m)$ and formula $\psi := \psi_1 \wedge \cdots \wedge \psi_k$. Clearly, $\psi$ is true in $\mathsf{G}(\omega)$ for any inference of $e_m$. But that means that we again have a "literal" learning. Otherwise, since $p$ is not 1, we will have a "fault" for sufficiently large $n$.

In (Blass et al., 1998), the authors proved a version of the zero-one law for binary sequences and, within the context, a decision problem. Our formal proof is a generalization to a class of algorithms in which logical inference admits a standard graph representation. Namely, we just proved the following:

**Theorem 1** *For almost all proofs, any learning algorithm of inference, based on randomization in $G(\omega)$, that necessitates veracity of inference, is almost surely literal learning.*
The complete formal proof is in C.2. Before, we established that there is a natural model for the inference and pointed out the limitations associated with it. In other words, the algorithm almost surely fail unless it is vacuous. ∎

**Theorem 2 (reformulation of the theorem 1)** Given the graph model of inference for machine learning, the only algorithm based on randomization, that also necessitates veracity of inference, is almost surely "literal" learning. In other words, for a sufficiently long proof, any algorithm that randomly deviates from the training data will fail with a probability of 1. ∎

---

[2]In a second-order logic, one can quantify over sets of domain elements; in the first-order logic, one can quantify over elements only.

**Corollary 3** Within a rigorous inference context, almost surely, no randomization of the prediction scheme of proof patterns can discover new (unknown) non-trivial valid statements.

This can be easily explained: since any degradation is inherited in the foundational graph, the subsequent inferences on the trained data tend to deviate from already shortened erroneous paths thus multiplying the faults. ∎

**Example of an inference problem that exceeds the current capabilities of generative learning**

*Elementary example*. Proving the statement: "for every number $2^n$ for any natural $n$, there exists a number $k$ such that $2^n * k$ does not have zeroes in its digital representation".

*Non-elementary example*. Dedekind numbers sequence.

*Benchmark Examples* Multiple benchmark examples can be found in Glazer (2024).

**Theorem 4 (Generalizations of main result)** Any algorithm of learning enforcing veracity, admitting a 0-1 domain cast as random graphs, is almost surely vacuous.

**Proof** This is the context where proof of theorem 1 is fully applicable. ∎

Within the view adopted herein, there is an interesting example of a decidable theory that admits a 0-1 random graph domain yet its classifier comparison is not expressible in its first-order logic. Therefore, it is an example of a.s. learning algorithm with randomization which is ultimately decidable but vacuous and does not support any notion of expressible first-order classifier comparison; thus, there is no feasible notion of fairness for classification tasks.

## 3.2 ELEMENTARY PROBLEMS

### 3.2.1 QUESTION

Question: Can one cut a scalene triangle into two congruent scalene triangles? Answer: Microsoft Bing Copilot: "Certainly! Let's explore how we can cut a scalene triangle into two congruent triangles". Then Copilot generates two methods to create the cut: angle bisector method, and perpendicular bisector method which would work only for isosceles triangles, completely ignoring the fact that the original triangle is scalene. The fault is that the bisectors will not divide the opposite side into two equal segments. So, the subsequent application of angle-angle-side and side-side-side postulates is invalid. However, Bing Copilot "insists" and suggests the question:" Can you cut any triangle into two congruent triangles?" The predictable Copilot's answer is now that any triangle can be, while referring to the very answer to the previous question as a given (one can only note that it looks "logical"). Needless to say the process would be easily repeated with all sorts of fallacious geometrical statements. If the user points out an occasional contradiction, the Copilot produces a loop or changes the subject.

Claude (Antropic): This was a different experiment where the author tried to "teach" Claude to solve the aforementioned elementary problem. It took a few trials before Claude arrived at a plausible reformulation of the problem. A very positive result was, despite an inability to present a complete rigorous proof, Claude came up with a plan for how to obtain the proof. However, after a few unsuccessful attempts to implement the plan, and a few homework sessions later, we agreed that reaching the point is beyond Claude's capabilities yet. With the three assistants, our experience with Claude was the most pleasant and sensible.

Google Gemini (Bard) The result is similar to Claude's. After a few clarifications and direct clues, Gemini produced the following in bold: "Therefore, I cannot confidently claim to have proven the statement about the impossibility of cutting a scalene triangle into two congruent scalene triangles". Note that it is, formally, a weaker statement than what is necessary to solve the problem asked.

Similar results were obtained for other chatbots, e.g., Perplexity.

Incidentally, the paper (Trinh et al., 2024) depicts good results on solving geometry problems of Olympiad's level. First, we have to note that, because the first-order theory of Euclid geometry is

```
import Mathlib.Data.Nat.PrimeFin
import Mathlib.Data.Nat.Factors
import Mathlib.Data.Set.Finite

theorem exists_infinite_primes (n : ℕ) : ∃ p, n ≤ p ∧ Nat.Prime p :=
let p := Nat.minFac (Nat.factorial n + 1)

  have f1 : Nat.factorial n + 1 ≠ 1 := ne_of_gt <| Nat.succ_lt_succ <| Nat.factorial_pos _
  have pp : Nat.Prime p := Nat.minFac_prime f1
  -- have ppc := Nat.Prime ↑pp
  -- have ppc := minFac_to_nat pp
  have np : n ≤ p :=
    le_of_not_ge fun h =>
      have h₁ : p | Nat.factorial n:= Nat.dvd_factorial (Nat.minFac_pos _) h
      have h₂ : p | 1 := (Nat.dvd_add_iff_right h₁).2 (Nat.minFac_dvd _)
      pp.not_dvd_one h₂
  ⟨p, np, pp⟩
```

elementary in the logical sense (decidable), the task is achievable by a universal algorithm since we can work in the decidable first-order theory of $\mathbb{R}$.

Since the output is natural language (rather than in a code for an automated prover, unlike in the approach of (Zheng et al., 2022)), it isn't easy to assess the solution's performance. Because this transformer is trained on synthetic data, and proofs are relatively short by nature of the problems involved, due to our main result, likely, the solution does not exceed a threshold of vacuous/literal learning overall. A more formalized approach is presented in (Krueger et al., 2021).

In (Nezhurina et al., 2024) are more examples of basic reasoning breakdown for foundational industrial models.

### 3.3 Non-elementary Problems

Above is an example of "hallucinating": a formal proof of the infinitude of prime numbers in Lean 3 or 4.

Here is the critical fragment of the proof where randomization played a key role:

```
{ by_contradiction ,
        have h1 : p | fact N := dvd_fact (min_fac_pos M) a ,
        have h2 : p | 1 := ( nat . dvd_add_iff_right h1 ) .
                mpr ( min_fac_dvd M) ,
        exact prime . not_dvd_one pp h2 },
  { exact pp }
```

This latest fragment renders the proof unusable. One correct version is placed above, which is unlikely to be found elsewhere (since we use an explicit "Nat." prefix).

In (Nguen & Sarah, 2022), the authors describe multiple patterns of software development that reflect erroneous or sub-optimal code generated by Copilot. This leads to an elevated code churn and downward pressure on code quality in $GitHub$. Another survey, (Kabir et al., 2024), shows that coding questions generate up to 50% of errors. A similar study is conducted in (Macmillan-Scott & Musolesi, 2024).

### 3.4 Discussion

**Discussion - Primary**

These results easily explain the phenomenon of "hallucinations" and brittleness of the GPT models in a rigorous context. It also means that LLMs is unlikely to discover any new mathematical result of sufficient strength.

**Discussion-Datasets** In (Gendron et al., 2023), is shown that the baseline dataset construction for rigorous learning needs to be a formal exercise. Consider the task of equation completion in which one has to predict a missing symbol. Since this is perfectly aligned with the main premise of LLM based on transformers, one can expect that the success rate for this task will be quite high. As is shown in the paper, this is not the case. An associated (and well-known) phenomenon of a plateau of performance and subsequent degradation in an exponential fashion manifests in the same way as for the generic sequence case. Similarly, few authors summarize a few problems in the answers in contemporary systems associated with a low P/R w.r.t citation usage from the underlying sources. These experimental results are not for the rigorous context.

There is a widespread belief that because the training set contains "everything", any result, including novel ones, can be proven using symbolic inference from the corpus. However, it is just not the case. It is well-known that any mathematical problem of significance requires one or multiple critical insights that are just not to be found. These are not combinations of known results (or tactics), but rather completely new, albeit inevitable, ideas. For instance, for some long-standing problems, new fields of mathematics had to be created, representing a new body of knowledge. Thus, generalizing the LLM solution for these targets is a task of yet another level of complexity for which the method is not suited. Moreover, as we show below, it is guaranteed to fail. The inevitable conclusion is that the apt inference model has to be more deferential to logic.

# A   MAIN RESULT

In this section, we assume a natural representation of a proof by a path from a node $e_0$ to the node $e_t$ in graph $G(\omega)$ in appendix B.1 which is, in a suitable enumeration, corresponds to a premise/target statements $e_0$, $e_t$ accordingly.

**lemma 1.** *(Accumulating errors Lemma). Assuming independence of faults in* $\mathsf{G}(\omega)$ *representing proofs, the probability of no fault proof tends to zero exponentially over its length.*

*Proof.* We can assume that faults are independent since the semantics of a formal inference are out of scope [3]. Let labeled graph $G$ represent proofs (chains) of enumerated statements (nodes) where each label is a probability that the chain ending with the corresponding node contains an error. Then we have:

$$\mathbb{P}(no\ fault\ proof) \leq exp(-\mathbb{E}(-number\ of\ faults)) \tag{2}$$

Since the right side of the equation tends to zero, we have:

$$\lim_{n \to \infty} \mathbb{P}(no\ fault\ proof\ of\ length\ n) = 0. \tag{3}$$

This proves the statement of the lemma.  ∎

**example 1.** Since GPT has no semantic notion on the entities involved, we can assume the lemma is fully applied to the GPT rigorous context.

**Corollary 4** Assume an algorithm admitting model $\mathsf{G}(\omega)$ for inference and using randomization. Then. the rate of (correctness) decay is exponential over the proof length.

**Proof** The proof is similar to a usual consideration for a set of independent events in a classic probability space generating a fault. The key observation is that, once a fault in the chain of inference occurs, it is thus erroneous in the chain subsequently. This proves that the correctness rate decays exponentially over the length of proof.  ∎

**(Heuristic note)** In a few papers, this phenomenon has been shown experimentally. Moreover, it has a few incarnations. These are hallucinations (when there are no references, supporting an inference), erroneous statements (falsehood, incorrect generalization, non sequitur, etc.), and a general misalignment. Exponential decay is also noted in a few papers; our result (Corollary 4) shows for all non-trivial (complex enough) tasks, including performance degradation on synthetic data in an autophagous loop.

**theorem 1.** *(Inherent* $\mathsf{GPT}$/LLM *Limitation).*

*Any algorithm of inference, based on randomization on* $\mathsf{G}(\omega)$*, that necessitates veracity of inference, is almost surely literal learning.*

*Proof.* (Informal) We give two proofs of the statement. To develop a theoretical intuition, we start with the one below. The second one, more instructive and rigorous, is in C.2.

Note that we can assume that one can enumerate all the inferences using $[n]$, since there is only a countable number of (finite) proofs on a countable number of entities (statements). Without loss of generality, for that representation of entities, we can assume that node (vertices) $e_i$ implies $e_j$ only if they are connected; we do not need to impose any order on the nodes.

For a generative model, that would be enumeration for a proof generated for a particular prompt, say, prove that $e_k$ implies $e_l$. Moreover, we can assume that a generic proof is an actual chain of thought, i.e., we have a finite sequence of distinct nodes, connected via regular paths, with possible cycles which would reflect the equivalency of the statements. The underlying training graph for this is not necessarily connected, but the model output has to contain a path from the premise to the desired conclusion.

---

[3]GPT algorithm does not follow the syntax of the first-order theory – instead, it uses randomizing and inferred statistics. It has no notion of non-statistical meaning.

In appendix B, we define the first-order language of graphs used in an associated model, $\mathsf{G}(\omega)$.

Thus, the path (or "chain of thought") is just a sequence of tuples $(s_k \sim s_l)$ where sign "$\sim$" represents adjacency for vertices $s_k$ and $s_l$ and there is a path

$$e_s \sim e_1 \wedge e_1 \sim e_2 \wedge \cdots \wedge e_k \sim e_t \tag{4}$$

Now, there are two possibilities:

1. The path (4) exists in the training set (not a novel context).

2. The path (4) does not exist in the training set (a novel context).

Consider the first-order formula $\phi(.) = e_s \sim e_1 \wedge e_1 \sim e_2 \wedge \cdots \wedge e_k \sim e_t$. Then, again, we have two possibilities. Namely, by 0-1 law (B.1), we have:

$$\lim_{n \to \infty} \mathbb{P}(\mathsf{G}(\omega) \models \phi) = 0 \text{ or } 1. \tag{5}$$

Thus, we have four possibilities, namely:

1. The limit (5) is equal to zero and the path (4) does not exist in the training set.

2. The limit (5) is equal to zero and the path (4) exists in the training set.

3. The limit (5) is equal to one and the path (4) does not exist in the training set.

4. The limit (5) is equal to one and the path (4) exist in the training set.

For each of these, we also need to consider the cases of model temperature, normalized to probability **p**, equal to zero or one, or between zero and one. We can assume the following for these cases:

For the case **1**, it is nearly obvious that, within the context, no valid proof can be found almost for sure in the first place either if we try literal learning, falling into a novel context, or varying the probability **p** between zero and one - we apply Accumulating errors Lemma since GPT is an accumulating errors algorithm. The latter manifests as a phenomenon of accumulating errors for sufficiently complex (lengthy) proofs.

The case **2** is more interesting. Despite having proof in the training set and a chance of literal learning, we use probability $p$ other than one. As a result, we are having the phenomenon of accumulating errors described above.

The case **3** is the most interesting – we are in a novel context – and may follow fragments of the proof, somehow creating the final proof as an assembly. Note, we chose $p$ equal to one. It means that we are trying to assemble the required proof in pieces. The problem is equivalent to finding paths among potentially connected pieces. However, we can simply apply B.1 and note that since $p$ is equal to 1, we have:

$$\lim_{n \to \infty} \mathbb{P}(G(n) \models \phi) = 1 \Leftrightarrow \mathsf{G}(\omega) \models \phi. \tag{6}$$

Therefore, for ever-growing complexity and length of proofs, we have to follow ever-growing fragments of proof literally which means we have them in the training set. That is literal learning or we have a contradiction with the assumption of this case.

The case **4** *is* literal learning, by definition.

The conclusion is that *almost for sure*, only literal learning, has a chance of generating an error-free proof.

$\lim_{n \to \infty} \mathbb{P}(G(n) \in \mathcal{A}) = 0$ or it is equal to 1. If it is zero, then no valid proof can be found within the context in the first place. Therefore, $\lim_{n \to \infty} \mathbb{P}(G(n) \in \mathcal{A}) = 1$. By Lemma 0, in the first-order theory

for the language of random graphs, it follows that $\mathsf{G}(\omega) \models \mathcal{A}$. For $p = 1$, it is possible. However, it also means that our inference follows a literal graph representation from the original (i.e., from the given training set). Thus, $p \neq 1$. In this case, we create a hierarchy in $\mathsf{G}(\omega)$ as follows.

Consider chain $(e_0 \xrightarrow{\psi_1} e_1 \ldots e_i \ldots e_j \xrightarrow{\psi_k} e_m)$ and formula $\psi := \psi_1 \wedge \cdots \wedge \psi_k$. Clearly, $\psi$ is true in $G_\omega(p)$ for any inference of $e_m$. But that means that we again have a "literal" learning. Otherwise, since $p$ is not 1, we will have a "fault" for sufficiently large $n$. $\blacksquare$

## B    FORMAL DEFINITION FOR LANGUAGE $L$

Let $L$ be a language (an extension of a basic formal logical language, $L_0$).

**Definition and base notations**    The set of $L-$terms is the smallest set $L_t$ such that contain all constant symbols of $L$, all variables, and if $t_1, t_2, ..., t_n$ are in $L_t$ then for any n-ary function symbol $f$, $f(t_1, t_2, ..., t_n)$ is also in $L_t$. Set $L_a$ of atomic formulas are represented by the properties:
(1) if $t_1$ and $t_2$ are terms then $t_1 = t_2$ is in $L_a$, and
(2) the corresponding n-ary function symbols are also in $L_a$.

In other words, the set of all formulas in $L$ (expressions, sentences - herein, we use these interchangeably) is the smallest set containing all atomic formulas and closed under logical connectives $\vee, \wedge, \neg, \rightarrow, \leftrightarrow$, quantifiers $\exists, \forall$, equality symbol " = ", parenthesis "(" and ")", and variables. For our purposes herein and simplicity, it is sufficient to consider that theory in language $L$ is a set of sentences in first-order logic over $L$. We also assume first-order logic with equality; in other words, only normal models are employed. Thus, the models, considered herein (e.g., Erdős–Rényi or finite graph model for random graphs, are normal).

The main language in this paper is that of graphs. [4] We denote $\mathbb{G}_L$ the first-order theory over language of graphs $L$. One convenient (and usual) laxity talking about expressions and formulas in $L$ is using $L$ and $\mathbb{G}_L$ interchangeably.

### B.1    0-1 LAW FOR GRAPHS $L$

We introduce a few known formulations for the 0-1 law for finite graphs.

**0-1 Lemma 0** For any first-order formula $\phi$ and graph $G$ in $\mathbb{G}_L$ (with the equivalent notation $\mathsf{G}(\omega)$ which is intuitively more suitable), let

$$G_{n,\phi} = \frac{|\{G \models \phi : |G| = n \text{ and } G \text{ is a graph}\}|}{|\{G : |G| = n \text{ and } G \text{ is a graph}\}|} \quad (7)$$

Then $\lim_{n \to \infty} G_{n,\phi}$ is $0$ or $1$.

*Proof.* Refer to, e.g., (Fagin, 1976).

This can be reformulated as
**0-1 Lemma** For any property $\mathcal{A}$ that can be described by a first-order expression $\phi$ and $G_n = \{G : |G| = n$ and $G$ is a graph$\}$,

$$\lim_{n \to \infty} \mathbb{P}(G_n \in \mathcal{A}) \in \{0, 1\} \quad (8)$$

To wit (assuming notations for $G(\omega)$, a set of all finite graphs, and its associated domain $\mathsf{G}(\omega)$, up to isomorphism):
**Lemma 0, reformulation** For any graph $G_n \in \mathsf{G}(\omega)$, $\lim_{n \to \infty} \mathbb{P}(G_n) = 0$ or $1$. The equivalent statement is as follows: for any first-order expression $\phi$ in theory of $\mathbb{G}_L$, $\lim_{n \to \infty} \mathbb{P}(G_n \models \phi) = 0$ or $1$. We can also say that $\lim_{n \to \infty} \mathbb{P}(G_n) \models \phi) = 1 \Leftrightarrow \mathsf{G}(\omega) \models \phi$.

**Lemma 0, reformulation** For any *random* graph $G_n \in \mathsf{G}(\omega)$, $\lim_{n \to \infty} \mathbb{P}(G_n) = 0$ or $1$. The equivalent statement is as follows: $\forall$ 0-1 probability $p$ and a first-order expression in theory of Random Graphs, $\phi$, $\lim_{n \to \infty} \mathbb{P}(G_n \models \phi) = 0$ or $1$. We can also say that $\lim_{n \to \infty} \mathbb{P}(G_n \models \phi) = 1 \Leftrightarrow \mathsf{G}(\omega) \models \phi$.

**Proof** Standard considerations similar to the previous lemma.     ∎
One useful representation for the same results is as follows. Given a first-order property $\mathcal{A}$ of a

---

[4]i.e. graph is a pair $G = (G, E)$ for non-empty set $G$ of nodes (vertices) and a binary relation $E$ on $G$ (the edges). For our purposes, we can assume that $G$ is symmetric and unordered: $E(a, b) \rightarrow E(b, a)$, and $E(a, a)$ is false. We denote $\mathsf{G}(\omega)$ the class of finite graphs and, loosely, the associated first-order logic model, described in the following section B.1.

random graph $G_n$, $\lim_{n \to \infty} \mathbb{P}(G_n \in \mathcal{A}) \in \{0, 1\}$. Equivalent notation will be $G(n, \omega)$ or just $G(n)$ when the context is clear.

## C    CORE PROOFS, PROOF OF THE MAIN THEOREM

**Lemma on** GPT. GPT prediction schema is (a.s.) an accumulating error algorithm unless it acts as a vacuous learning.

### C.1    FIRST PROOF (FORMAL)

**Proof** Suppose the GPT prediction is not a vacuous/literal learning. Consider formula $\phi = \wedge_i \phi_i$. where $\phi_i$ are respected edges on a proof sequence paths, $e_i \xrightarrow{\phi_i} e_j$, in any enumeration of the nodes in the training dataset. In our first-order theory of graphs, this is a first-order expression. Moreover, since the theory obeys 0-1 law, for inference graph $G$, by Lemma B.1, $\lim_{n \to \infty} G_{n,\phi}$ is zero or one. Since GPT algorithm is not vacuous/literal learning, we have $\lim_{n \to \infty} G_{n,\phi} = 0$. That means for any $\epsilon$ there exists $n_0$ such that $n >= n_0$ implies $\lim_{n \to \infty} G_{n,\phi_n} < \epsilon$. Viewed as a graph in $\mathsf{G}(\omega)$ and GPT randomization with temperature $p$ selected for inference, the graph satisfies conditions of 1. Thus, starting from $n_0$, GPT must be an (a.s.) catastrophic / accumulating error algorithm, with the veracity of proof exponentially tending to zero. That is to say, it has to be almost surely a vacuous learning to generate valid proof.

### C.2    SECOND PROOF (FORMAL)

**Main Theorem** *For almost all proofs, any learning algorithm of inference, based on randomization in $G(\omega)$, that necessitates veracity of inference, is almost surely literal learning.*

**Proof** More instructive than informal considerations is the following proof in which we partially follow a version of the 0-1 law in (Blass et al., 1998).

The probability space for the GPT algorithm can be viewed as follows. Consider a probability distribution over infinite binary strings. Let $\Psi$ be a set of infinite sequences representing proofs (since any string can be encoded by a binary string, in a suitable enumeration (or embedding) and, given a proposition, its proofs of any length can be encoded into an infinite binary string).

Let $\Psi$ be a set of infinite sequences $\phi = \langle \phi_n : n \geq 1 \rangle \in \Psi$. In this context, we can view the set as one of independent trials. The resulting probability distribution over $\Psi$ is naturally equipped with the product measure (cf. (Feller, 1968)). Moreover, we can consider every proof over strings semantically. Therefore, for any generative algorithm $\mathfrak{A}$, if, given a sequence $\{e_0 \to e_1 \ldots e_k \to e_t\}$, representing the proof $\{e_0 \to e_t\}$, we have $\mathfrak{A}(\phi_n) = e_n$, we say that the algorithm succeeds proving $\phi_n$; otherwise, we say it fails. The corresponding notation for any $\phi \in \Phi$, if $\mathfrak{A}$ succeeds, is $\mathfrak{A} \models \phi$; if $\mathfrak{A}$ fails, we write $\mathfrak{A} \not\models \phi$.

Thus, let us introduce the notation: $p_n(\mathfrak{A}) = \mathbb{P}(\mathfrak{A} \text{ fails on the n-th step } \phi_n \text{ of } \phi)$ or $\mathbb{P}(\mathfrak{A} \not\models \phi_n)$ where $\phi$ ranges over $\Psi$.

The following two cases are possible:

**Case 1**. There exists an algorithm, $\mathfrak{A}$ s.t. $\sum_{n=0}^{\infty} p_n(\mathfrak{A}) < \infty$. By the (first) Borel-Cantelli lemma (Feller, 1968), $\mathbb{P}(\text{there are infinitely many } n \text{ s.t. } \mathfrak{A} \text{ fails on } \phi_n) = 0$. Thus, for almost all $\phi \in \Psi$, $\mathfrak{A}$ succeeds on all but finitely many $\phi_n$. Therefore, for almost all $\phi$, there exists an algorithm $\mathfrak{A}' = \mathfrak{A} +$ finite lookup that succeeds on $\phi$. The algorithm $\mathfrak{A}$ stays the same for all $\phi$ and only the finite lookup depends on $\phi$. It means that, for almost all sequences $\phi \in \Psi$,

$$\mathbb{P}(\mathfrak{A} \models \phi) = 1. \tag{9}$$

The question becomes whether such an algorithm $\mathfrak{A}$ can be GPT. We will show below that the assumption it is GPT meets a contradiction. Namely, from (9) we have:

$$\forall \epsilon > 0 \ \exists n_0 > 0 \ s.t. \ \forall n > n_0 \ \mathbb{P}(\mathfrak{A} \models \phi_n) > 1 - \epsilon. \tag{10}$$

On the other hand, from the Accumulating error lemma inequality (2), we see that $\mathbb{P}(\mathfrak{A} \not\models \phi) > 1 - exp(-\rho)$ where $\rho = \mathbb{P}(\mathbb{E}(\#faults))$. Thus, setting $\epsilon = 1 - exp(-\rho)$ leads to contradiction with (10). This leaves only two possibilities for the algorithm $\mathfrak{A}$ to succeed (since we have $\mathbb{P}(\mathfrak{A} \models \phi) = 1$ for all $\phi$).

In the first instance, $\mathfrak{A}$ may arrive at nodes representing the false statements, but the inferences would be true (vacuous truths). The proof is still invalid, overall. The second instance is literal learning; that is, the algorithm would generate (potentially, piece-by-piece) a known proof discoverable in the training data.

**Case 2**. For every algorithm $\mathfrak{A}$, $\sum_{n=0}^{\infty} p_n(\mathfrak{A}) = \infty$. Again, as in (2), we can assume that $\phi_n$ are independent events. By the (second) Borel-Cantelli lemma (e.g., (Feller, 1968)), the probability that there exists an infinite number of $n$ that $\mathfrak{A}$ fails on $\phi_n$ is 1. Hence, for every $\mathfrak{A}$ there exists $n$ s.t. $\mathbb{P}(\mathfrak{A} \models \phi_n) = 0$. Since there are only countably many algorithms, for almost all $\phi \in \Phi$, we have:

$$\mathbb{P}(\exists \mathfrak{A}, \ \mathfrak{A} \models \phi) = 0. \tag{11}$$

Qualitatively, this means that in this case, almost surely, no algorithm using randomization with exponential correctness decay can succeed in generating a proof for the statement. ∎

**Main Theorem, Reformulation** *For almost all proofs, any learning algorithm of inference, based on randomization in* $\mathsf{G}(\omega)$*, does not generate a valid proof unless it is vacuous.* ∎

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
