# OpenReview forum: "On inherent limitations of GPT/LLM \\ Architecture"
_ICLR.cc/2025/Conference — Submitted to ICLR 2025_

### Official Review · Reviewer_r172 · 2024-10-31

**Soundness:** 1
**Presentation:** 1
**Contribution:** 1
**Rating:** 1
**Confidence:** 3

**Summary:**

The paper presents theoretical results demonstrating the limitations of GPT/LLM architecture for reasoning.

**Strengths:**

It is a nice topic.

**Weaknesses:**

The presentation e.g. the theorem and proof on lines 167-171 is unconvincing.

**Questions:**

Is this a draft? It seems like the presentation is strange, at the very least quite unfamiliar in style to me.

---

### Official Review · Reviewer_NbbF · 2024-10-31

**Soundness:** 1
**Presentation:** 1
**Contribution:** 1
**Rating:** 1
**Confidence:** 4

**Summary:**

The main manifest of the paper is that LLM such as GPT are inherently incapable in the problem of producing rigorous proofs, build on chain of reasoning, unless such a proof has been exactly observed in the training set. It is also mentioned that this fundamental impossibility is due to the attention mechanism, and/or due to the finite-temperature prediction of the architecture, meaning that the next output token is not always the most likely token.

The proof is based on ideas from first-order logic, which I am not familiar with, but if I understand correctly, goes something like this: Since the proof requires a chain of reasoning comprised of multiple connections, and since each such connection there is a probability that it will not be sampled correctly, then overall the probability of the LLM to output the correct proof decays exponentially with the length.

**Strengths:**

1) The question whether LLMs can produce rigorous proofs or not is fascinating, and at the center of scientific interest.
2) The logic-based model is potentially an interesting model to address such questions.

**Weaknesses:**

1) There is an empirical evidence that LLMs have some capability in providing rigorous proofs, even if currently a weak one. The paper does not seem to address this evidence too much (except the Trinh et al 2024 paper on line 229), and explain why its reasoning is not applicable in those cases. Much more detailed comparison is required.

2) Section 2.1 presents the first order-logic model, and the main theorems. I am not familiar with first-order logic (as said), but this model appear to be general, and not tightly connected to neither any specific machine learning algorithm, nor LLMs or GPT. Theorem 1 appears colloquial – what does it rigorously mean “necessitates veracity of inference”? How is “literal learning” is defined? To establish the statement targeted by the authors, I think that a much more detailed model is required. Afterwards, there are examples of inference problems the are not solvable by LLMs. These are just mentioned without any justification.

2) Experiments are lacking: Section 2.2 pretenses a couple of “question and answer” examples, with a few models, and it is claim that “Calude was the most pleasant and sensible”. This part is not scientific. These are just simple experimentation withe models and general impression.

3) Section 2.3: It is impossible for me to understand this example from the displayed code.

4) The concluding paragraph following – “There is a widespread belief that because the training set” – reads more like a personal opinion rather than a solid conclusion from the paper.

5) Proofs: These are difficult to follow because there is no methodological structure of accurate definitions of terms, follows by a clear definition of the probabilistic model, before the proofs. For a quick read, there is no technical challenge in the proofs, and these are just a description in mathematical terms of the ideas that explained in the paper in words (except for the use of Borell-Cantelli lemmas to prove 0-1 laws)

6) While it is occasionally mentioned that the attention mechanism of GPT is an inherent cause for its impossibility to provide novel rigorous proofs, this is not actually discussed in the paper.

**Questions:**

Here are a few minor and representation comments:

1) The paragraphs staring in line 69 almost repeats the one starting in line 64.
2) The sentence “The logical view approach is more effective in generalization by varying a domain obeying the generic 0-1 law.” is an example to a vague sentence.
3) Before Equation (1), “and $e_t$ respectively” – something is missing here.
4) In line 155, what is $p$ ? It is never defined.
5) Line 189 – “Generalization of the main result Theorem 5” – where is Theorem 5 ? There are many such inconsistencies.
6) Line 389 – Isn't it the other way round between possibilities 1 and 2 – the first is not a novel context and the second is ?

---

### Official Review · Reviewer_YQxe · 2024-10-31

**Soundness:** 2
**Presentation:** 1
**Contribution:** 2
**Rating:** 3
**Confidence:** 3

**Summary:**

This paper investigates the reasoning abilities of large language models (LLMs), examining their potential limitations in producing truly novel and rigorous proofs. The authors argue that while these models can mimic logical structures and provide plausible responses, they lack the capacity for original, rigorous proof generation.

**Strengths:**

The problem of assessing reasoning capabilities of LLMs versus memorization is very relevant to the current machine learning community.

**Weaknesses:**

- The paper seems to be not polished: in the introduction there is a repeated paragraph (lines 69-73), several concepts (e.g. literal learning, first order model graph, generic 0-1 law, hallucination) should be defined formally or explained better, there is no literature review paragraph, the flow is hard to follow. Section 2.1 cannot be followed without reading the appendix, a re-organization of the flow would be helpful.
- The formatting in Section 2.1 is not clear: Q.E.D in line 171 does not seem to be the end of the proof of Theorem 1, and it is not mentioned where the proof of Theorem 1 can be found. Same for lines 176 and 191. In Corollary 3 it is not clear where the statement ends.
- Because the lack of formalism in the presentation, the results seem unconvincing in this version of the paper.

**Questions:**

- You denote a finite graph $G(\omega)$, what is $\omega$ exactly?
- what is the difference between $C(e_0, e_t) $ and $e_0 \sim e_t$?
- Line 150: why do we have only these 2 possibilities for the probability of $G$ satisfying property $A$?
- Line 153: it is not clear where to find Lemma 0. A pointer to the exact subsection of the appendix would be useful, or you can move it to the main (the main is shorter than the page limit).
- Line 155: what is p? Line 157: what are $\psi_i$?
- It is not clear where the proofs of the results in Section 2.1 can be found.
- Section 2.2.1: Can you explain more in details the experiments that you run (e.g. prompts, models used, number of experiments for each model-prompt, answers received, metric to validate correctness of answers).
- Line 227: `similar results were obtained for other chatbots, e.g. Perplexity': can you be more specific on those?
- Line 229: Can you explain better why the results from Trinh et al.,2024 on geometry problems are not a contraddiction to your framework?
- Can you explain the code in page 6?
- In the introduction and abstract the authors mention that the inability of LLMs of producing original proofs follows from their architecture. However, I do not see any discussion of e.g. the self-attention mechanism in the main contributions. Could you please expand on this?

---

### Official Review · Reviewer_dypc · 2024-11-04

**Soundness:** 1
**Presentation:** 1
**Contribution:** 1
**Rating:** 3
**Confidence:** 4

**Summary:**

The paper studies the reason why today’s LLM models struggle with multi-hop reasoning tasks and produce hallucinations in the process. It presents an argument for why the fundamental nature of inference on the transformer architecture prevents them from being able to generate correct mathematical proofs or output false statements in other settings which require rigor.
To do so, the paper considers the setting of first-order predicate statements on random graphs.
Under this model, the paper claims that a GPT model has exponentially decreasing probability of finding the correct proof unless it has already seen the proof in the train set.

**Strengths:**

- The paper aims to understand fundamental limitations of autoregressive next-token predictors which is an important problem today for ML.

**Weaknesses:**

-	The presentation in the paper severely lacks a level of formalism necessary a technical conference. The writing is at times vague and this makes it hard to verify or critique the main Theorems and Corollaries presented. Some of the issues with the formalism are listed in the questions section below.
-	Theorem 1 seems to be wrong (?) It is well-known that GPT/LLM models can generalize and produce novel proofs so the statement that for almost all proofs, the model is performing literal learning appears to be incorrect?
-	The paper relies heavily on “randomization in inference” of GPT/LLM models. This randomization is not clearly quantified or defined anywhere which makes it hard to follow the argument.
-	Moreover, there are many post-training techniques one could employ to increase the chances of producing correct output. The paper’s claim seems to potentially hold only in the limit where the size of reasoning chains tend to infinity and we wish to be correct over all possible novel statements on the graph.

**Questions:**

1. If auto-regressive next-token predictors are universal (Malach 2023),  how do you re-concile the limitation supposedly proved in this work?


Issues with formalism:
1. What is a GPT model? It needs to be formally defined.
2. Where is Lemma 0? Why is it in the Appendix and not referred to in line 150?
3. In line 150, the appropriate result which allows us to conclude that the limiting probability is either 0 or 1 (Fagin’s 0-1 law) needs to be cited.
4. How are we excluding the limiting probability being 0 without having described what G(w) is?
5. What does it mean that “our inference follows a literal graph representation from the original” and why is it implied in line 154?
6. What is p in line 155?
7. What is your formal definition of “learning model for inference”?
8. Do the limitations proposed go away under greedy decoding rather than temperature based sampling?
9. In Lemma 1, what is the definition of a “fault” in G(w)?
10. It is not made clear where exactly the randomness in GPT generations is being used. Note that randomness alone doesn’t mean a non-zero probability of an error in a single step. We can have a probability distribution that always assigns probabilities of 0 or 1 for different tokens.

Typos/formatting errors:
- Lines 68-73 the paragraph is repeated

---

### Meta-Review · Area_Chair_YkXq · 2024-12-23

**Metareview:**

The paper attempts to prove inherent limitations of GPT/LLM-based architectures for reasoning and theorem proving by invoking arguments from first-order logic and 0-1 laws on random finite graphs. While the topic is relevant and timely—understanding the fundamental reasoning constraints of autoregressive large language models is indeed a critical research question—unfortunately, the submission in its current form does not meet the quality standards for acceptance.

**Additional Comments On Reviewer Discussion:**

1.	Lack of Clear Formalism and Rigor:
Across all reviews, a central recurring critique is the lack of a clean, formal, and rigorous presentation. Key concepts such as “literal learning,” the exact nature of the probability space, the role of randomization, and the definitions of the models and proofs are not articulated with sufficient clarity. The core theorems are stated vaguely, and multiple reviewers struggled to parse their precise meaning or verify their soundness. Terms like “necessitates veracity of inference” and the formal definition of the GPT/LLM model in the logical framework are not adequately introduced.
	2.	Unconvincing or Incomplete Theoretical Claims:
The main theorem asserts that with high probability, the model either memorizes known proofs or fails to produce correct new proofs as complexity grows. Reviewers noted that from a practical standpoint, LLMs have demonstrated some ability to produce proofs not explicitly seen in the training set. The paper does not convincingly reconcile these observed capabilities with the theoretical claims. Moreover, the paper’s reliance on “randomization” is not clearly quantified, and how it applies to inference (e.g., temperature sampling) versus other decoding strategies (e.g., greedy decoding) is insufficiently discussed.
	3.	Insufficient Connection to Empirical Evidence and Current Techniques:
While the paper mentions known difficulties of LLMs in reasoning tasks, it provides only a few anecdotal examples without a rigorous experimental setup. There is no thorough comparison with existing results or known methods that mitigate reasoning errors. Post-training interventions like fine-tuning, chain-of-thought prompting, retrieval-augmented generation, or proof verifiers are not addressed in detail, leaving a gap between the theoretical claims and current practical progress in the field.
	4.	Presentation and Structural Issues:
The manuscript contains repeated paragraphs, unclear references to theorems and lemmas, and confusing formatting. Lemmas and main results are referenced without proper pointers, and some theorems (e.g., Theorem 1, Theorem 5) appear without easily accessible or well-structured proofs. The reviewers found the paper difficult to read and interpret due to these presentation problems.

Conclusion:

All four reviewers concluded that the paper, as it stands, is not acceptable for publication. The authors’ responses acknowledge some of the issues and indicate plans for a substantial revision. However, given the current version’s shortcomings in formality, clarity, and empirical grounding, as well as the reviewers’ consistent recommendation to reject, the final decision is to reject.

The authors are encouraged to restructure the paper with more explicit definitions, clearer statements of results, rigorous proofs, and careful integration with existing literature and known results. Addressing these points could strengthen the paper’s impact and credibility for future submission attempts.

---

### Decision · Program_Chairs · 2025-01-22

Reject